# Corona virus fear among health workers during the early phase of pandemic response in Nepal: A web-based cross-sectional study

Pratik Khanal[1], Kiran Paudel[1,2]*, Navin Devkota[3], Minakshi Dahal[4], Shiva Raj Mishra[5], Devavrat Joshi[3]

**1** Institute of Medicine, Tribhuvan University, Kathmandu, Nepal, **2** Nepal Health Frontiers, Kathmandu, Nepal, **3** National Academy for Medical Sciences, Kathmandu, Nepal, **4** Center for Research on Environment, Health and Population Activities (CREHPA), Kathmandu, Nepal, **5** Nepal Development Society, Chitwan, Nepal

\* kiranpaudel59@gmail.com

## Abstract

Health workers involved in the COVID-19 response might be at risk of developing fear and psychological distress. The study aimed to identify factors associated with COVID-19 fear among health workers in Nepal during the early phase of the pandemic. A web-based survey was conducted in April-May 2020 among 475 health workers directly involved in COVID-19 management. The Fear Scale of COVID 19 (FCV-19S) was used to measure the status of fear. Multivariable logistic regression was performed to identify factors associated with COVID fear. The presence of COVID-19 fear was moderately correlated with anxiety and depression, and weakly correlated with insomnia. Nurses, health workers experiencing stigma, working in affected district, and presence of family members with chronic diseases were significantly associated with higher odds of developing COVID-19 fear. Based on the study findings, it is recommended to improve the work environment to reduce fear among health workers, employ stigma reduction interventions, and ensure personal and family support for those having family members with chronic diseases.

## Introduction

The psychological implications as a result of disease outbreaks are often neglected by the health system [1–3], although studies have found that the proportion of mental health effects is higher than the effect of a particular disease during epidemics [4]. One of the emotions involved in mental health outcomes in people during disease outbreaks is fear. It is an adaptive defense mechanism which when become chronic can lead to adverse mental health effects [5, 6]. The progressive nature and scientific uncertainties related to infectious diseases create fear among people especially when the infection and death rate is alarming [7].

The onslaught of COVID 19 is currently burdening the health systems and paralyzing economies across the world. Nepal, a South-Asian country, ranking low in health security index (111 out of 195 countries) [8] is not an exception from the threat of COVID-19. The country

**Data Availability Statement:** The raw data in the form of tables has been uploaded as supporting information.

**Funding:** The author(s) received no specific funding for this work.

**Competing interests:** The authors have declared that no competing interests exist.

**Abbreviations:** AOR, adjusted odds ratio; CI, confidence interval; COVID-19, Corona virus 2019; FCV-19S, Fear of COVID-19 Scale; HADS, Hospital Anxiety Depression Scale; ISI, Insomnia severity index; PPE, personal protective equipment; SARS, Severe Acute Respiratory Syndrome.

reported its first case on January 23, 2020 [9, 10], and the total infection toll reached 8,13,011 cases along with 11416 deaths as of November 1, 2021 [11]. The increasing rate of infection is putting a strain on its already compromised health system [12]. Health care workers who are at the frontline of managing COVID 19 are prone to developing adverse mental health outcomes during this situation. They are likely to develop fear attributed to their close and longer interaction with suspected patients, a better understanding of disease development, and its progression [13]. Early evidence has shown increased work pressure, inadequate protective measures, risk of infection, and transmission of infection to family members, limited organizational support and exhaustion contributing to adverse mental outcomes including fear in health care workers [3, 14–16]. Fear and stress experienced by health workers affect their work, behaviour and health outcomes [3, 17].

Nepal is a low-and middle-income country whose health system is constrained by inadequate human resources (0.7 doctor and 3.1 nurses per 1000 population) [18], low health system preparedness to disasters and health emergencies, weak coordination between the federal, province and local governments, limited diagnostic facilities, and lack of critical health care resources [19]. The limited health system capacity and poor quarantine management by local administration might have added burden on health workers including propagation of COVID-19 fear [20]. Further, Nepal's health system in the recent past has been regularly affected by disasters such as Gorkha Earthquake 2015, Indo-Nepal border closure, outbreaks of infectious diseases, floods, and landslides among others. COVID-19 outbreak and its consequences including lockdown, loss of economy, disruption in regular health services, lack of critical resources including oxygen and protective equipment and inadequate financial motivation among health workers have further crippled the country's health system capacity including its health workers [21].

The understanding of fear and other psychological outcomes among health workers has not received much attention during the early phase of the pandemic. Limited studies so far have investigated the mental health impact of COVID-19 among health workers in Nepal. In this regard, this study aims to assess the status of COVID-19 fear among health workers involved in the COVID-19 response in Nepal. In addition, this study aims to explore the relationship of COVID-19 fear with other mental health outcomes among health workers.

## Materials and methods

### Study design, participants, and procedures

A total of 475 health workers participated in the study. A web-based cross-sectional survey was conducted among health workers directly involved in COVID-19 management between April 26 and May 12 in 2020. Social media groups of professional organizations were identified, and health workers were requested for their interest in participating in the study. Study objectives were explained in the google forms, and e-informed consent was taken from all the participants before the data collection. Those health workers who expressed interest were personally invited to fill up the Google forms. Health workers were required to provide their valid email address for quality control purposes. The inclusion criteria for the study were those aged 18 years and above, currently working in Nepal, and involved in the COVID-19 response. The study protocol was approved by the Ethical Review Board of the Nepal Health Research Council (Registration number: 2192; 315/2020). Details about the recruitment and the demographic details of the study participants have been mentioned in an article published elsewhere [22].

### Measures

The fear scale of COVID 19 (FCV-19S) was used in the study to assess the fear among health workers. It is a relatively new scale developed in 2020 [3] and has been used in different

countries including India [13], Bangladesh [23], Israel [24], Italy [25], Turkey [26] and Eastern Europe [27]. The FCV-19S has seven items and five-point Likert scales ranging from 1 to 5, with lower and higher values indicating strongly disagree and strongly agree, respectively. The total scores range between 7 and 35, and higher the score, higher the fear of COVID-19. Similarly, the 14-item Hospital Anxiety and Depression Scale (HADS) was used for measuring anxiety (HADS-A, 7 items) and depression (HADS-D, 7 items), and the 7-item Insomnia Severity Index (ISI) was used for measuring insomnia.

Sociodemographic information included age (up to 40, >40 years), gender (male, female), ethnicity (*Brahmin/Chhetri*, *Janajati* and others), educational qualification (Intermediate and below, bachelor, and masters and above), marital status (single, ever married), family type (nuclear and joint), profession (doctors, nurses, others), living with children (yes, no), living with older adults (yes, no), presence of chronic disease among family members (yes, no) and history of medication for mental health problems (yes, no). Similarly, job-related variables included type of health facility (primary and, secondary and tertiary), work experience (up to 5 and >5 years), work role in COVID-19 response (frontline, second line), adequacy of precautionary measures in the work place, (not sufficient, sufficient), awareness of government incentives for health workers (yes, no), perceived stigma (yes, no, do not want to answer), working in the affected district (yes, no) working overtime (yes, no) and change in regular job duty during COVID-19 (yes, no). Working in the affected district was defined as a district with at least one case during the time of data collection.

## Data analysis

The sociodemographic and job related characteristics, and itemwise response of the FCV-19S are presented as frequencies and percentages. The pattern of the relationship between FCV-19S and other psychometric tools (HADS-A, HADS-D and ISI) was explored by calculating the correlation coefficient (S2 Table). The outcome variable was not normally distributed so for analysis purposes, the median value obtained from FCV-19S was calculated and those having scored more than the median (>16) were categorized as the presence of fear and less than or equal to median as an absence of fear of COVID-19. A chi-square test was performed between categorical independent and categorical dependent variables (S3 Table) and those variables significant at the 10% significance level were fitted in the multivariable logistic regression model [28]. In the regression model, the effects of gender, ethnicity, profession, education, working in the affected district, family member with chronic disease, faced stigma, precautionary measures in the workplace, awareness about government incentive and history of medication for mental health problems were adjusted. Strength of the association was determined by an adjusted odd ratio (AOR) at 95% confidence interval (CI). One of the independent variables, history of medication for mental health problems was also fitted into the model although it was not significant in the bivariate analysis as it was supposed to alter psychological outcomes [29]. The variance inflation factor (VIF) was measured before conducting multivariable logistic regression analysis which did not detect multicollinearity (VIF value less than 1.3).

## Results

### Socio-demographic and job- related characteristics of health workers

Table 1 shows the sociodemographic and job-related characteristics of health workers. Among 475 health workers, 52.6% were female and 65.9% belonged to the *Brahmin/Chhetri* ethnic group. The professional category comprised of nurses (35.2%), doctors (33.9%), paramedics (17.9%) and other health professionals (13%). Likewise, 25.1% were living with children, 34.3% were living with an elderly, 54.5% had a family member with a chronic medical

**Table 1. Sociodemographic and job-related characteristics of health workers.**

| Variables | Category | N (%) | Variables | Category | N (%) |
|---|---|---|---|---|---|
| Age (years) | | | Living with elderly (>60 years) | | |
| | 20–29 | 325 (68.4) | | Yes | 163 (34.3) |
| | 30–39 | 124 (26.1) | | No | 312 (65.7) |
| | 40–49 | 19 (4.0) | Family member with a chronic medical condition | | |
| | 50 and above | 7 (1.5) | | Yes | 259 (54.5) |
| | Mean age in years (±SD) | 28.20 (±5.80) | | No | 216 (45.5) |
| Sex | | | History of medication for mental health | | |
| | Male | 225 (47.4) | | Yes | 22 (4.6) |
| | Female | 250 (52.6) | | No | 453 (95.4) |
| Ethnicity | | | Type of health facility | | |
| | Brahmin/Chhetri | 313 (65.9) | | Primary | 84 (17.7) |
| | Janjati | 110 (23.2) | | Secondary and tertiary | 391 (82.3) |
| | Madheshi | 52 (6.1) | Work role | | |
| | Dalit | 7 (1.5) | | Front line | 215 (45.3) |
| | Others | 16 (3.4) | | Second line | 260 (54.7) |
| Education | | | Work experience (years) | | |
| | Intermediate and below | 94 (19.8) | | Up to 5 | 336 (70.7) |
| | Bachelors | 277 (58.3) | | >5 | 139 (29.3) |
| | Masters and above | 104 (21.9) | Precautionary measures in the workplace | | |
| Position | | | | Sufficient | 100 (21.1) |
| | Nurse | 167 (35.2) | | Not sufficient | 375 (78.9) |
| | Doctor | 161 (33.9) | Experience of stigma due to occupation | | |
| | Paramedics | 81 (17.1) | | Yes | 255 (53.7) |
| | Public health professional | 32 (6.7) | | No | 199 (41.9) |
| | Laboratory staff | 19 (4.0) | | Do not want to answer | 21 (4.4) |
| | Pharmacist | 15 (3.2) | Aware of government incentives for health workers | | |
| Marital status | | | | Yes | 270 (56.8) |
| | Single | 299 (62.9) | | No | 205 (43.2) |
| | Ever married | 176 (37.1) | Change in regular job duties during COVID-19 | | |
| Family type | | | | Yes | 334 (70.3) |
| | Nuclear | 308 (64.8) | | No | 141 (29.7) |
| | Joint | 167 (35.2) | Working overtime during COVID-19 | | |
| Living with children | | | | Yes | 233 (49.1) |
| | Yes | 119 (25.1) | | No | 242 (50.9) |
| | No | 356 (74.9) | | | |

condition and 4.6% had a history of medication for mental health problems. The majority of the health workers in this study (82.3%) worked in either secondary or tertiary level health facilities. The proportion of health workers reporting insufficient precautionary measures in the workplace, facing stigma, awareness of government incentives for health workers, change in job duties during COVID-19 and working overtime was 78.9%, 53.7%, 56.8%, 70.3% and 49.1% respectively.

## Itemwise distribution of response of FCV-19S

Table 2 shows the itemwise distribution of response of FCV-19S. The proportion of health workers who either strongly agree or agree to the individual items of FCV-19S was highest

**Table 2. Itemwise distribution of responses.**

| Scale | Items | Strongly disagree | Disagree | Neither agree nor disagree | Agree | Strongly agree |
|---|---|---|---|---|---|---|
| | | N (%) | N (%) | N (%) | N (%) | N (%) |
| FCV-19 S1 | I am most afraid of corona virus disease-19 | 65 (13.7) | 150 (31.6) | 132 (27.8) | 103 (21.7) | 25 (5.3) |
| FCV-19 S2 | It makes me uncomfortable to think about corona | 80 (16.8) | 177 (37.3) | 84 (17.7) | 122(25.7) | 12 (2.5) |
| FCV-19 S2 | My hands become clammy when I think about corona | 159 (33.5) | 196 (41.3) | 74 (15.6) | 40 (8.4) | 6 (1.3) |
| FCV-19 S4 | I am afraid of losing my life because of corona | - | 304 (64.0) | 77 (16.2) | 80 (16.8) | 14 (2.9) |
| FCV-19 S5 | When watching news and stories about corona on social media, I become nervous and anxious | 87 (18.3) | 150 (31.6) | 84 (17.7) | 129 (27.2) | 25 (5.3) |
| FCV-19 S6 | I cannot sleep because I am worrying about getting Corona | - | 372 (78.3) | 68 (14.3) | 31(6.5) | 4 (0.8) |
| FCV-19 S7 | My heart races or palpitates when I think about getting corona | 147(30.9) | 192 (40.4) | 76 (16.0) | 48 (10.1) | 12 (2.5) |

(32.5%) for 'When watching news and stories about corona on social media, I become nervous and anxious' and lowest (7.3%) for 'I cannot sleep because I am worrying about getting Corona'. The descriptive analysis of the items of the FCV-19S is shown in S1 Table.

**Correlation of FCV-19 S with HADS-A, HADS-D and ISI.** The correlation analysis showed that FCV-19S was moderately correlated with HADS-A (r = 0.513, p<0.001) and HADS-D (r = 0.425, p<0.001) but weakly correlated with ISI (*r* = 0.367, p<0.001). The seven items of the FCV-19S were either weakly or moderately correlated with HADS-A, HADS-D and ISI (p<0.001) (S2 Table).

**Predictors of COVID-19 fear among health workers.** In the adjusted analysis, profession, stigma experience, working in the affected district and having family members with chronic diseases were significantly associated with COVID fear. Compared to other health workers, nurses (AOR = 2.29; 95% CI: 1.23–4.26) had significantly higher odds of having COVID fear. Similarly, health workers working in the affected district (AOR = 1.76; 95% CI: 1.12–2.77), those having family members with chronic diseases (AOR = 1.50; 95% CI: 1.01–2.25), and those who faced stigma (AOR = 1.83; 95% CI: 1.12–2.73) had significantly higher odds of having COVID fear compared to those not working in affected district, not having a family member with chronic disease, and those not facing stigma respectively. Gender, ethnicity, education, precautionary measures, awareness about government incentives, and history of medication for mental health problems were however not statistically significant with COVID fear (Table 3).

## Discussion

This study documents the factors associated with the presence of fear related to COVID-19 among health workers in Nepal in the early phase of the pandemic. The study identified profession, working in the affected region, presence of family member with chronic disease and stigma faced by health workers as significant factors contributing to the presence of COVID fear among health workers. In this study, nurses had significantly higher odds of having COVID fear than other health workers. This might be because of their role in providing patient care more closely, frequently and for a longer duration compared to other health workers. The chance of being infected and transmitting infection to others, dealing with highly infective disease and uniqueness of the cases might have led to increased fear among nurses. Similar findings were noted in studies conducted in other countries that have reported COVID-19 cases and countries that have handled epidemics such as Severe Acute Respiratory Syndrome (SARS) in the past [14, 29–32]. Effective strategies to reduce fear with a focus on nurses are thus required to avert COVID fear and psychological distress which might include

**Table 3. Factors associated with COVID related fear among health workers (n = 475).**

| Variables | Categories | Fear N (%) | Unadjusted OR (95% CI) | Adjusted OR (95% CI) |
|---|---|---|---|---|
| **Gender** | | | | |
| | Male | 83 (37.9) | Ref | Ref |
| | Female | 136 (62.1) | 2.04 (1.41–2.95)* | 1.15 (0.66–1.99) |
| **Ethnicity** | | | | |
| | Brahmin/Chhetri | 129 (58.9) | Ref | Ref |
| | Janajati | 68 (31.1) | 2.02 (1.31–3.12)* | 1.56 (0.97–2.51) |
| | Madheshi | 11 (5.0) | 0.87 (0.40–1.91) | 1.04 (0.45–2.39) |
| | Others | 11 (5.0) | 2.62 (0.94–7.25) | 1.84 (0.61–5.49) |
| **Profession** | | | | |
| | Doctor | 55 (25.1) | 0.78 (0.49–1.24) | 0.78 (0.46–1.32) |
| | Nurses | 106 (48.4) | 2.64 (1.67–4.17)* | 2.29 (1.23–4.26)* |
| | Others | 58 (26.5) | Ref | Ref |
| **Education** | | | | |
| | Intermediate and below | 53 (24.2) | Ref | |
| | Bachelor | 128 (58.4) | 0.67 (0.42–1.06) | 0.83 (0.49–1.41) |
| | Masters and above | 38 (17.4) | 0.45 (0.25–0.79) * | 0.77 (0.39–1.52) |
| **Affected district** | | | | |
| | Yes | 174 (79.5) | 1.76 (1.15–2.68)* | 1.76 (1.12–2.77)* |
| | No | 45 (20.5) | Ref | Ref |
| **Family member with chronic disease** | | | | |
| | Yes | 132 (60.3) | 1.54 (1.07–2.22)* | 1.50 (1.01–2.25)* |
| | No | 87 (39.7) | Ref | Ref |
| **Precautionary measures** | | | | |
| | Sufficient | 37 (16.9) | Ref | Ref |
| | Insufficient | 182 (83.1) | 1.61 (1.02–2.53) * | 1.49 (0.91–2.45) |
| **Faced stigma** | | | | |
| | Yes | 136 (62.1) | 1.89 (1.31–2.72) * | 1.83 (1.12–2.73)* |
| | No | 83 (37.9) | Ref | Ref |
| **Aware about government incentive** | | | | |
| | Yes | 112 (51.1) | 0.65 (0.45–0.94)* | 0.79 (0.53–1.19) |
| | No | 107 (48.9) | Ref | Ref |
| **History of medication** | | | | |
| | Yes | 7 (3.2) | 0.53 (0.21–1.33) | 0.60 (0.23–1.58) |
| | No | 212 (96.8) | Ref | Ref |

*Statistically significant at p<0.05.

support from the management especially with incentives, job rotation, and working hours, family and social support, psychological first aid, adequate personal protective equipment, and regular capacity building activities among others [32–34].In our study, more than half of the health workers experienced stigma during COVID-19 pandemic. Stigma faced by health workers was also significantly associated with higher odds of the presence of fear of COVID-19. Already vulnerable due to exposure to possible infections, emotional exhaustion due to increasing workload, deployment to newer settings such as fever clinics and lack of adequate PPEs, health workers are more likely to face stigma either internalized or from public, which will impair their performance in the COVID-19 response [35]. Stigma reduction strategies should thus be employed for educating the public through mass media and community

engagement activities [36, 37]. Equally important is to identify the underlying causes of stigma experienced by health works during the epidemic.

Working in the affected district was significantly associated with the presence of fear among health workers. This is obvious as they need to directly deal with COVID-19 patients or those at risk of infection. Health workers in Hubei Province of China [14] during the COVID pandemic and health workers directly involved in the care of patients in Canada [38] during the SARS epidemic also faced more psychological distress compared to those not involved in the direct care of COVID patients or less affected areas. As fear among health workers reflects psychological wellbeing, health workers working in risk districts should be supported emotionally and due attention is required on their workload and safety needs.

In this study, the presence of family members with chronic diseases had higher odds of the presence of COVID-19 fear. The fear of transmission to family members and the vulnerability posed by chronic disease conditions might have resulted in a higher degree of COVID fear among health workers. This finding is similar to the study from China [39] where health workers were concerned with the infection of their family members. Personal and family support is thus required for health workers having family members with chronic diseases.

## Conclusion

Our study findings showed that COVID fear was moderately correlated with anxiety and depression, suggesting a detrimental effect of COVID fear on psychological well-being. Perhaps, symptoms of anxiety and depression were a consequence of working in a fearful environment for an extended period. Health facility managers need to monitor the psychological well-being of their staff and ensure proper psychological intervention measures are adopted in a timely and precise manner. In this study, only one out of five health workers mentioned protective measures in their workplace as sufficient. Similarly, just over half of health workers were aware of the government incentives entitled to them during COVID-19. This reflects the need to improve organizational and policy aspects for boosting the work morale of health workers to reduce fear and psychological distress among health workers involved in the COVID-19 response.

## Strength and limitation

Our study has some limitations to be noted. This study was conducted during the early phase of the pandemic in Nepal when fewer than 300 COVID-19 cases were reported. The status of fear thus might have altered thereafter. Further follow-up studies might be required among health workers to understand the changes in psychological outcome such as fear. Similarly, participation in this study required internet access and survey was administered in English language. This might have left out health workers who did not have internet access and had difficulty in comprehending English language. Likewise, health workers who were in fear due to COVID-19 pandemic might not have participated in the study. Also, the results might have been affected by the subjective response. Despite limitations, this study identifies those at risk of developing fear using a new scale. The evidence generated can be useful to those at the decision-making level and health facility managers for designing appropriate interventions to enhance psychological well-being among health workers in current and similar epidemics in the future.

## Supporting information

**S1 Table. Descriptive analysis of the items of the English version FCV-19S.**
(DOCX)

**S2 Table. Correlation of FCV-19 S with HADS-A, HADS-D and ISI (N = 475).**
(DOCX)

**S3 Table. Fear of COVID-19 and its associated factors.**
(DOCX)

## Author Contributions

**Conceptualization:** Pratik Khanal, Kiran Paudel, Navin Devkota.

**Data curation:** Pratik Khanal, Kiran Paudel, Minakshi Dahal.

**Formal analysis:** Pratik Khanal, Minakshi Dahal, Shiva Raj Mishra.

**Investigation:** Pratik Khanal, Kiran Paudel, Navin Devkota.

**Methodology:** Pratik Khanal, Kiran Paudel, Navin Devkota, Shiva Raj Mishra.

**Project administration:** Pratik Khanal.

**Resources:** Pratik Khanal, Devavrat Joshi.

**Supervision:** Shiva Raj Mishra, Devavrat Joshi.

**Validation:** Pratik Khanal.

**Writing – original draft:** Pratik Khanal, Minakshi Dahal.

**Writing – review & editing:** Pratik Khanal, Kiran Paudel, Navin Devkota, Minakshi Dahal, Shiva Raj Mishra, Devavrat Joshi.

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
