## [Decision Letter · Decision Letter 0]

23 Sep 2021

 PGPH-D-21-00428 Corona virus fear among health workers during the early phase of pandemic response in Nepal: a web-based cross-sectional study PLOS Global Public Health

Dear Dr. Paudel,

Thank you for submitting your manuscript to PLOS Global Public Health. After careful consideration, we feel that it has merit but does not fully meet PLOS Global Public Health’s publication criteria as it currently stands. Therefore, we invite you to submit a revised version of the manuscript that addresses the points raised during the review process.

 We require that you address the concerns of reviewer 2 before we proceed with a final decision.  

We look forward to receiving your revised manuscript.

Kind regards,

Vaibhav Saria, Ph.D

Academic Editor

Journal Requirements:

1. In your Methods section, please provide additional information about the participant recruitment method and the demographic details of your participants. Please ensure you have provided sufficient details to replicate the analyses such as: 

a) a description of any inclusion/exclusion criteria that were applied to participant recruitment, 

b) a statement as to whether your sample can be considered representative of a larger population, and 

c) a more detailed description of how participants were recruited.

2. During our internal evaluation of the manuscript, we found text overlap between your submission and the following previously published works:

https://www.grin.com/document/1007212

Please revise the manuscript to rephrase the duplicated text and cite your sources.

3. Thank you for providing consent information for your study. However, we note that you have not provided ethical approval information. We understand that the framework for ethical oversight requirements for studies of this type may differ depending on the setting and we would appreciate some further clarification regarding your research. Could you please provide further details on why your study is exempt from the need for approval and confirmation from your institutional review board or research ethics committee (e.g., in the form of a letter or email correspondence) that ethics review was not necessary for this study? Please include a copy of the correspondence as an "Other" file.

4. In the online submission form, you indicated that "Data can be make available after asking with first and corresponding author in reasonable request.". All PLOS journals now require all data underlying the findings described in their manuscript to be freely available to other researchers, either 1. In a public repository, 2. Within the manuscript itself, or 3. Uploaded as supplementary information.

5. We have noticed that you have uploaded supporting information but you have not included a list of legends.  Please add a full list of legends for all supporting information files (including figures, table and data files) after the references list.

Additional Editor Comments (if provided):

Thank you for your submission, we are very glad to invite you to resubmit your revised manuscript after it addresses the questions raised by the reviewers. In particular, please address the concern of the reviewer regarding PLOS's Data availability statement you can find here- https://journals.plos.org/globalpublichealth/s/data-availability.

Reviewers' comments:

Reviewer's Responses to Questions

**Comments to the Author**

1. Does this manuscript meet PLOS Global Public Health’s publication criteria? Is the manuscript technically sound, and do the data support the conclusions? The manuscript must describe methodologically and ethically rigorous research with conclusions that are appropriately drawn based on the data presented.

Reviewer #1: Yes

Reviewer #2: Yes

2. Has the statistical analysis been performed appropriately and rigorously?

Reviewer #1: Yes

Reviewer #2: I don't know

3. Have the authors made all data underlying the findings in their manuscript fully available (please refer to the Data Availability Statement at the start of the manuscript PDF file)?

Reviewer #1: Yes

Reviewer #2: No

4. Is the manuscript presented in an intelligible fashion and written in standard English?

Reviewer #1: Yes

Reviewer #2: Yes

5. Review Comments to the Author

Reviewer #1: Dear Author

The topic seems to be relevant with the current pandemic situation. It may be helpful for policy implication and reference for future research. However, abstract and introduction seems to have little information. Could you please add some references for this article? Other factors remain excellent.

Thank you

Reviewer #2: - Perhaps consider bulking up your introduction with additional Nepal-specific information: have there been anything else in the literature regarding fear not necessarily due to COVID but due to other crises, other infectious diseases? Being able to make that comparison (recognizing that COVID-19 is of much much larger proportion), might allow us to make some inference (even if not causal) on the magnitude of fear this time around.

- You might want to consider using a multi-pronged approach for your recruitment methods (for next time). For example, people who are anxious or depressed or very fearful might be taking some time off social media - this limits the people who are responding to your survey to those who might not be feeling the extreme side of things. How else could you reach them? You can keep it web-based, but perhaps finding a listserv of emails from the professional organizations and sending them emails might be another opportunity to reach a wider range of people. (And people might be more willing to participate on a survey vs social media.)

- How did you calculate your sample size? Why 475? What did the 475 participants represent in terms of representativeness of the country as a whole - were they working in urban/rural settings, were they older/younger, what type of facility? Who do these 475 participants represent? How did you classify “being involved in the COVID-19 response” - working directly with COVID-19 patients? Or working in a hospital in other departments? Working in research regarding the COVID-19 response?

- Something else to think about in terms of your population is that some might be triggered given their ongoing experience with COVID-19 and might not want to participate in this study - these limitations should be mentioned.

- For your logistic regression - it’s not necessary to rely on significance level (is there a reasoning for the 10%?) to create your model (in fact, it’s not recommended). Instead, you should use the literature to determine which variables should be included and accounted for. For example, has “awareness of government incentives” shown to determine fear? If so, based on your expertise and the literature, then yes, including it is great, if not, you might want to rethink its inclusion.

- In terms of health care workers, are community health care workers not included?

- You briefly mentioned this in the conclusion, but talking a bit more about the health system shortages in Nepal (lack of PPE, lack of widely available vaccines etc.) could also have contributed to the fear.

- You also mentioned “effective strategies to reduce fear with a focus on nurses” - what are some potential recommendations (based on the literature) of strategies that can be helpful here?

6. PLOS authors have the option to publish the peer review history of their article (what does this mean?). If published, this will include your full peer review and any attached files.

**Do you want your identity to be public for this peer review?** For information about this choice, including consent withdrawal, please see our Privacy Policy.

Reviewer #1: No

Reviewer #2: No

---

## [Editor Report · Decision Letter 1]

22 Nov 2021

Corona virus fear among health workers during the early phase of pandemic response in Nepal: a web-based cross-sectional study

PGPH-D-21-00428R1

Dear Dr. Paudel,

We're pleased to inform you that your manuscript has been judged scientifically suitable for publication and will be formally accepted for publication once it meets all outstanding technical requirements.

Within one week, you'll receive an e-mail detailing the required amendments. When these have been addressed, you'll receive a formal acceptance letter and your manuscript will be scheduled for publication.

An invoice for payment will follow shortly after the formal acceptance. To ensure an efficient process, please log into Editorial Manager at https://www.editorialmanager.com/pgph/ click the 'Update My Information' link at the top of the page, and double check that your user information is up-to-date. If you have any billing related questions, please contact our Author Billing department directly at authorbilling@plos.org.

Kind regards,

Vaibhav Saria, Ph.D

Academic Editor